

# An anti-collusion attack defense method for physical layer key generation scheme based on transmission delay

Xiaowen Wang[1], Jie Huang[1,2], Chunyang Qi[1], Yang Peng[1] and Shuaishuai Zhang[1]

[1] Southeast University, Nanjing, China
[2] Purple Mountain Laboratories, Nanjing, China

## ABSTRACT

Physical layer security (PLS) is considered one of the most promising solutions to solve the security problems of massive Internet of Things (IoTs) devices because of its lightweight and high efficiency. Significantly, the recent physical layer key generation (PLKG) scheme based on transmission delay proposed by Huang et al. (2021) does not have any restrictions on communication methods and can extend the traditional physical layer security based on wireless channels to the whole Internet scene. However, the secret-sharing strategy adopted in this scheme has hidden dangers of collusion attack, which may lead to security problems such as information tampering and privacy disclosure. By establishing a probability model, this article quantitatively analyzes the relationship between the number of malicious collusion nodes and the probability of key exposure, which proves the existence of this security problem. In order to solve the problem of collusion attack in Huang et al.'s scheme, this article proposes an anti-collusion attack defense method, which minimizes the influence of collusion attack on key security by optimizing parameters including the number of the middle forwarding nodes, the random forwarding times, the time delay measurement times and the out-of-control rate of forwarding nodes. Finally, based on the game model, we prove that the defense method proposed in this article can reduce the risk of key leakage to zero under the scenario of the "Careless Defender" and "Cautious Defender" respectively.

## INTRODUCTION

With the development of communication and Internet of Things technology, more and more intelligent devices are used in all aspects of society and life. Urban intelligent transportation, industrial modernization, smart grid, smart home, and smart driving all need the support of intelligent devices (*Asghari, Rahmani & Javadi, 2019*). Large-scale heterogeneous devices are faced with increasingly complex application scenarios and increasingly blurred network boundaries (*Kouicem, Bouabdallah & Lakhlef, 2018*), and more and more application scenarios require cross-domain information interaction. However, due to the lack of effective encryption communication scheme support, unencrypted sensitive data can be easily intercepted by third parties, resulting in serious

Corresponding author
Jie Huang, jhuang@seu.edu.cn

security problems such as information leakage and data tampering (*Lu & Xu, 2019*), which may seriously endanger the safety of people's lives and property.

Faced with these large-scale heterogeneous devices especially IoT devices with security requirements, traditional identity authentication based on digital certificates and physical layer key generation (PLKG) schemes based on asymmetric keys (*Harn & Ren, 2011*) face the problems of high cost, difficult key management, and inapplicability to devices with limited resources. However, recent studies (*Lee, Hwang & Choi, 2020*; *Aldaghri & Mahdavifar, 2020*; *Tang et al., 2021b*) have shown that the characteristics of communication devices, channels, and noise can be skillfully used in the physical layer to realize device identity identification, authentication and key distribution without complicated mathematical operations and key management.

Physical layer security technology is a supplement to cryptography security technology. In radio frequency communication, it is mainly used to improve the security performance of wireless communication networks. For example, the wireless channel-based physical layer key generation (*Kai et al., 2010*; *Aldaghri & Mahdavifar, 2018*; *Li et al., 2017*) schemes are to use the uniqueness and reciprocity of wireless channel characteristics to generate keys, including received signal strength (RSS), channel impulse response in the time-frequency domain, phase, delay, envelope and other characteristics of the received signal. However, the wireless physical layer security technology cannot be applied to communication systems other than radio frequency communication, such as visible light communication (*Lopez-Martinez, Gomez & Garrido-Balsells, 2015*), underwater acoustic communication (*Xu, Fan & Liu, 2020*) and wired communication (*Salem et al., 2016*), which has certain limitations.

Therefore, researchers are looking for more general physical layer features to meet the security requirements of different scenarios. Recently, *Huang et al. (2021)* proposed a new physical layer key generation scheme based on network transmission delay, which used random forwarding technology and a three-stage delay measurement method to generate random and reciprocal communication delay and then converted the random delay into a secret key by means of quantization coding and information reconciliation. This scheme uses the characteristics of the network itself, which has no restrictions on the communication mode of nodes. Therefore, the physical layer key generation scheme based on network transmission delay greatly expands the use scenario of physical layer security and makes it possible to realize cross-domain key agreement.

However, when analyzing the security of Huang et al.'s scheme, we find that the security of the scheme is based on the secret apportionment strategy, that is, the secret information corresponding to the key is divided into multiple fragments and randomly apportioned to the nodes involved in forwarding. These nodes transmit the secret and finally recombine it at the receiving end so that only the two parties who negotiate the key can generate the key, while the nodes involved in forwarding can only master part of the secret but cannot generate the key. Therefore, there is a hidden danger of a collusion attack when the secret apportionment strategy is adopted. Once a number of malicious nodes share secrets, it is possible to steal the keys shared by both parties. We will discuss this in detail in Section 3.

To solve the security problem of collusion attacks in *Huang et al.*'s *(2021)* scheme, we propose an anti-collusion attack defense method, which can effectively reduce the risk of key leakage caused by a collusion attack. The contributions of this article are summarized as follows.

- Through the analysis of Huang et al.'s scheme, we find that the scheme has the security problem of collusion attack, and we reveal the influence of collusion attack by a single malicious node and multiple malicious nodes on key security by a probability model.
- In order to solve the collusion attack problem in Huang et al.'s scheme, we propose a defense strategy based on optimized deployment parameters to minimize the impact of collusion attacks, including the number of middle forwarding nodes, the random forwarding times, the number of delay measurement times and the level of out-of-control rate of middle forwarding nodes.
- Based on the game model, the optimal attack strategies of the "Careless Defender" and "Cautious Defender" are analyzed respectively. It is theoretically proved that the defense strategy proposed in this article can reduce the key security risk caused by collusion attacks to zero under the optimal attack strategy.

The remainder of the article is organized as follows. In Section 2, we briefly introduce the working principle and main components of Huang et al.'s scheme. In Section 3, we analyze the scheme of Huang et al. and put forward that the main security problem is the collusion attack. In Section 4, we give a defense strategy against collusion attacks and prove the effectiveness of this method. In Section 5, we give a literature review to give readers a comprehensive and systematic understanding of PLKG. Finally, the conclusion is drawn in Section 6.

## BACKGROUND

### Four properties of network physical features used to generate keys

Physical layer security research (*Jiao et al., 2019*; *Zeng, 2015*; *Wallace & Sharma, 2010*) shows that wireless channel features have unique reciprocity. If the sender and the receiver negotiate with the channel state information as the key, there is no need to distribute and manage the key, and secret communication can be carried out directly. The same for network physical features, a network feature $X$ that can be used for a key agreement should meet the following four properties:

- **Measurability** Any node $Z$ in the network can obtain the numerical network feature $X$ through network measurement. If it is continuous, it is marked as $X_Z(t)$, and if it is discrete, it is marked as $X_Z(n)$.
- **Randomness** $X_Z(t), X_Z(n)$ should be a stationary stochastic process. This is required to satisfy:

(a) The mathematical expectation is independent of time $t$, that is
$E[X_Z(t)] = E[X_Z(t + \tau)]$;

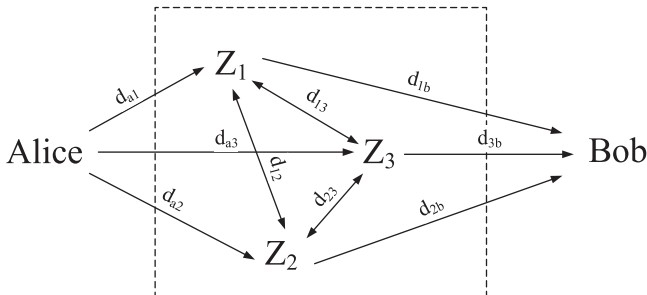

**Figure 1  Random forwarding network ($m = 3$).**

(b) The correlation function only depends on the time interval $\tau$, that is $R_{X_Z}(t, t+\tau) = R_{X_Z}(\tau)$. $X_Z(n)$ is the same.

- **Reciprocity** The network features obtained by both communication parties at the same time or the same measurement are approximately the same, that is, $X_A(t) \approx X_B(t), X_A(n) \approx X_B(n)$.
- **Entanglement** means that the reciprocity only belongs to two or more communicating parties, not to the network $G$ itself, that is, $\nexists G(t), \forall Z \in G, X_Z(t) \approx G(t)$, where $G(t)$ is the overall network feature. $X_Z(n)$ is the same.

Therefore, any network characteristics including transmission delay, bandwidth, throughput, bandwidth utilization, packet loss rate, network traffic, *etc.* (*Brownlee & Claffy, 2004*) can be used to generate the key as long as they meet the above four properties.

### *Huang et al.*'s *(2021)* physical layer key generation scheme based on transmission delay

Huang et al. first proved that the network transmission delay is an excellent network feature for key generation and proposed a practical key generation scheme. The main components of Huang et al.'s scheme are as follows.

#### *Random forwarding networks (RFNs)*

Huang et al.'s method utilize RFNs as the random source of measured delay. RFNs refers to a kind of network composed of several middle forwarding nodes which use random forwarding as their forwarding strategy. These middle forwarding nodes can be any devices connected to each other. Figure 1 shows a random forwarding network composed of three middle forwarding nodes.

The random forwarding rule is simple. Firstly, Alice randomly selects a middle forwarding node to send a delay measurement data packet, and sets the forwarding times as $N$ in the packet; Then, the middle forwarding node receiving the delay measurement data packet randomly selects one middle forwarding node in the RFN including itself as the next hop, and reduces the remaining forwarding times by 1; Finally, when the remaining forwarding times return to 0, the delay measurement data packet is directly sent to Bob.

**Peer**J Computer Science

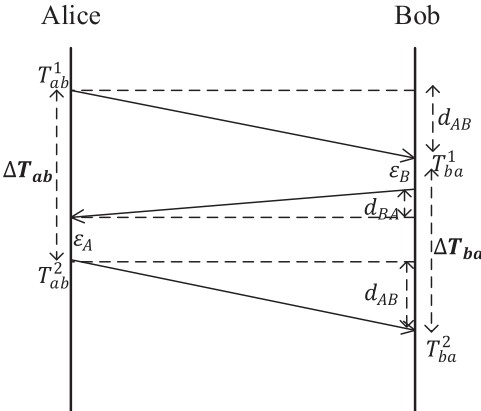

**Figure 2 Three-stage delay measurement protocol.**

Because the destination of each forwarding is random, the forwarding route from Alice to Bob is random, thus ensuring that the end-to-end delay of the delay measurement data packet sent from Alice to Bob through the random forwarding network is random.

The randomness of end-to-end delay is related to the number of middle forwarding nodes, forwarding times, and random forwarding strategy. We find the optimal random forwarding strategy to maximize the randomness of end-to-end delay and have proved that with the increase of the number of middle forwarding nodes and the number of forwarding times, the randomness of end-to-end delay is increasing (*Wang et al., 2022*).

### *Delay measurement protocol*

To ensure the reciprocity of the measured delay, the three-stage delay measurement protocol as shown in Fig. 2 is adopted.

The idea of a three-stage delay measurement protocol is as follows:

1. Alice sends a request delay measurement data packet through a random forwarding network to Bob according to the random forwarding rule and records the sending time $T_{ab}^1$.
2. Bob records the receiving time $T_{ba}^1$ when receiving the request delay measurement data packet, and sends a reply delay measurement data packet to Alice according to the random forwarding rule, assuming that the stay delay is $\varepsilon_B$.
3. Alice receives the reply delay measurement data packet and sends the final delay measurement data packet to Bob according to the forwarding route generated by the random forwarding network in i), assuming that the stay time is $\varepsilon_A$. Then Alice records the receiving time $T_{ab}^2$ and calculates the measured delay $\Delta T_{ab}$.
4. Bob receives the final delay measurement data packet and records the receiving time $T_{ba}^2$. Then Bob calculates the measured delay $\Delta T_{ba}$.

The measured delays of Alice and Bob are $\Delta T_{ab} = T_{ab}^2 - T_{ab}^1$ and $\Delta T_{ba} = T_{ba}^2 - T_{ba}^1$, respectively. Since the random route Alice forwarded to Bob twice is the same, then we have

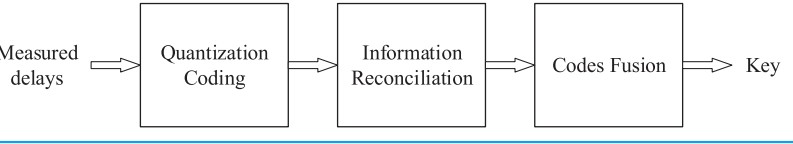

**Figure 3 Key generation process.**

$$\Delta T_{ab} = \Delta T_{ba} = d_{AB} + d_{BA} + \varepsilon_A + \varepsilon_B = T_{measured} \qquad (1)$$

where $T_{measured}$ is the secret shared by Alice and Bob. Obviously, with the support of RFNs and the three-stage delay measurement protocol, the measured delay satisfies all four necessary properties of network features that can be used for key agreement: measurability, randomness, reciprocity, and entanglement. Therefore, $T_{measured}$ can be used to generate keys.

It is worth mentioning that although the three-stage delay measurement protocol guarantees the reciprocity of measured delay in principle, in fact, due to the random fluctuation of the network, there will still be some small errors in the delay measured by Alice and Bob, which will be eliminated during key generation.

### Key generation process

As Fig. 3 shows, similar to the method of wireless physical layer key generation, first Alice and Bob respectively quantized and coded the measured delay into binary codes, then Alice and Bob corrected the different bits in the generated binary codes by means of information reconciliation (*Tang et al., 2021a*), and finally, Alice and Bob fused a group of binary codes generated by measured delay into a key, where codes fusion usually combines several short keys into one long key by random out-of-order splicing, so that the code generated by each delay can spread to the whole key, thus achieving the effect of resisting local exhaustive attacks.

## THE SECURITY PROBLEM OF HUANG ET AL.'S SCHEME

By the scheme of *Huang et al. (2021)*, Alice and Bob, the key negotiation parties, independently measured the reciprocal network characteristics and shared the random secret of measured delays. However, Alice's secret information can not be transmitted to Bob out of thin air.

In fact, Fig. 4 reveals the secret of keeping the consistency of the three-stage delay measurement protocol. From Fig. 4, it can be found that the middle forwarding nodes participating in this random forwarding can also obtain the same measured delay as Alice and Bob. From the communication point of view, it seems that Alice transmitted the random secret of measured delay to Bob through a random forwarding route, so the middle forwarding nodes carrying the task of transmitting measured delay naturally obtained this random secret. We call this feature **Secret Transmission Consistency**. It should be noted that although the communication point of view is used as an analogy, it is actually quite different from the plaintext secret transmission, so it is impossible to obtain the measured delay by monitoring the communication link.

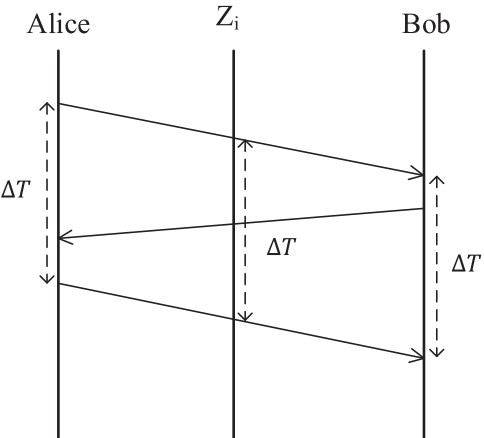

**Figure 4 Secret transmission consistency of the three-stage delay measurement protocol.**

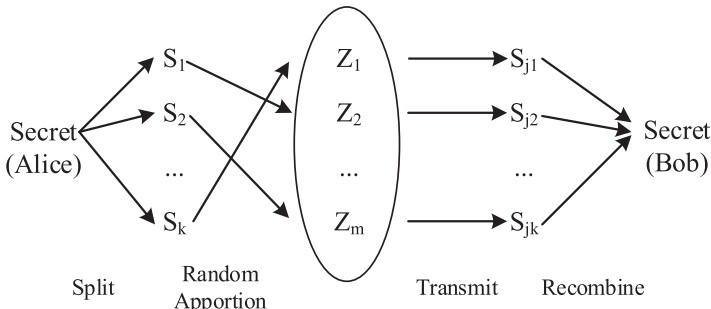

**Figure 5 Secret apportionment strategy.**

The key to Alice and Bob's secure communication is generated by a set of measured delays. As is shown in Fig. 5, if we regard the key as a complete secret shared by Alice with Bob, each measured delay is one piece of the whole secret. As the middle forwarding nodes are randomly selected by random forwarding rules, these secret fragments are randomly apportioned to all middle forwarding nodes. Only Alice and Bob can obtain the complete secret used to generate the key. As long as there are enough middle forwarding nodes, the possibility of obtaining a complete secret by one middle forwarding node is small enough to ensure the security of the key. This security strategy that relies on the number of middle forwarding nodes to apportion secrets is **Secret Apportionment Strategy**, which is the security mechanism of *Huang et al.*'s *(2021)* method.

However, the Secret Apportionment Strategy is most concerned with the **Collusive Attack**. A Collusive Attack refers to multiple malicious nodes conspiring to destroy the security of the target system. In our key agreement scenario, the Collusive Attack is manifested as malicious nodes conspiring to steal the key negotiated by both communication parties.

Collusive Attack has a great influence on the security of the Secret Apportionment Strategy. As shown in Fig. 6, once the attacker has mastered the vast majority of middle

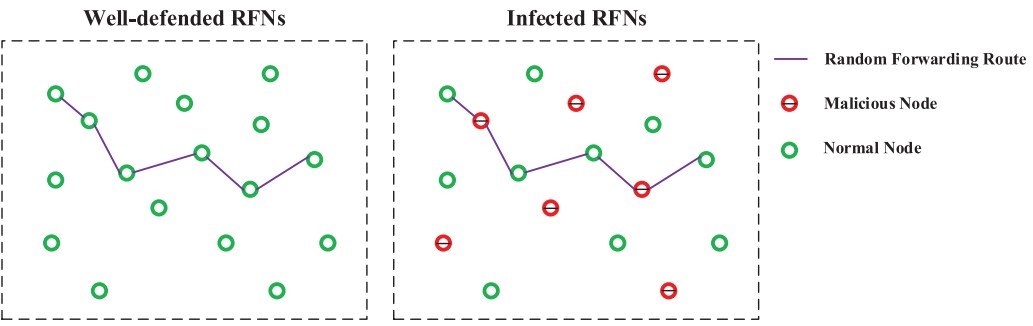

**Figure 6 Schematic diagram of a well-defended network and infection network.**

forwarding nodes in the RFNs, the key agreement process is almost completely monitored by the attacker. In this case, the key can be easily leaked.

Therefore, it is necessary to study the influence of the Collusive Attack on our method. The **Attack Model** of Collusive Attack under Secret Apportionment Strategy is given below:

1. Because the middle forwarding nodes have the same status in RFNs, it is assumed that the attacker's selection of attack targets is random.

2. Because the defense strategy is deployed on the middle forwarding node, attacking a middle forwarding node does not mean controlling the middle forwarding node. It is reasonably assumed that the attacker may fail to control the middle forwarding node that was attacked.

3. Because of the similar defense strategies deployed on the different middle forwarding nodes, it is assumed that the probabilities of the attacker successfully controlling the attacked middle forwarding nodes are the same.

The **Formal Description** of the problem of Secret Apportionment Strategy against the Collusive Attack is as followed:

Assuming that the number of middle forwarding nodes is $m$ and the number of forwarding times is $N$, this key is composed of $k$ measured delays. Then the composition of this key is shown in Fig. 7A, in which each time delay is determined by a random forwarding path $Z_{j1} \rightarrow Z_{j2} \rightarrow \ldots \rightarrow Z_{jN}$, and any forwarding node $Z_{ji}$ in the random forwarding path is randomly selected from the middle forwarding nodes $Z_1, Z_2, \ldots, Z_m$. If there exist malicious nodes in the random forwarding path, the delay corresponding to the random forwarding path is no longer secure. When all the delays are insecure, the key will be exposed as shown in Fig. 7B.

1) Considering the single malicious node, the probability of a single malicious node participating in a certain time delay measurement is $1 - (1 - 1/m)^N$. When $m$ is large, the probability of a single malicious node participating in one delay measurement is approximately $N/m$, and the probability of participating in all delay measurements is $(N/m)^k$. When the value of $k$ is slightly larger, this probability will be close to 0. Therefore, under the Secret Apportionment Strategy, a single malicious node can hardly pose a security risk to the key.

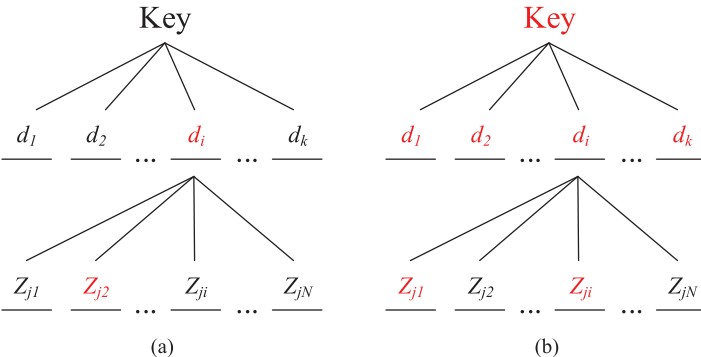

**Figure 7 Key composing structure and schematic diagram of Collusive Attack (color RED means being controlled by the attacker).**

2) Considering multiple malicious nodes, let us suppose that there are $r$ $(1 \leq r \leq m)$ malicious nodes conspiring to attack. According to the Secret Transmission Consistency of the three-stage delay measurement protocol, for one-time delay measurement, as long as a malicious node participates, the delay will be exposed to this malicious node. According to Fig. 7, the probability of time delay exposure is $1 - (1 - r/m)^N$. For a key agreement, the key is composed of $k$ measured delays, and the key will be exposed only when all these measured delays are exposed. Therefore, under the condition of collusion of $r$ malicious nodes, the key exposure probability $p_{expose}$ is

$$p_{expose}(r) = \left[ 1 - \left( 1 - \frac{r}{m} \right)^N \right]^k \tag{2}$$

Figure 8 shows the change curve of key exposure probability $p_{expose}$ with the collusion number $r$ of malicious nodes when $m = 100$, $N = 10$, and $k = 10$.

As can be seen from Fig. 8, there are two threshold numbers with significant curvature changes on the curve: $r_{defense}$ and $r_{attack}$. When $r < r_{defense}$, the key is basically secure; When $r_{defense} \leq r < r_{attack}$, the probability of key exposure $p_{expose}$ increases rapidly; When $r > r_{attack}$, the key exposure probability is close to the maximum value, and basically no longer increases. Because $r_{defense}$ indicates the number of malicious nodes that defenders can tolerate, We call $r_{defense}$ as *the upper limit of the defender's tolerance*. $r_{attack}$ indicates the lowest number of malicious nodes that can effectively destroy key privacy, so we call $r_{attack}$ as *the ideal lower limit of the number of malicious nodes controlled by the attacker*.

These two indicators, $r_{defense}$ and $r_{attack}$, are very important for key security. To explore the relationship between $r_{defense}$, $r_{attack}$ and $m$, $N$, $k$, the mathematical definition of $r_{defense}$ and $r_{attack}$ are described as follows:

$$\begin{cases} r_{defense} \overset{\Delta}{=} \underset{r}{\arg\max}\, K(p_{expose}) \\ r_{attack} \overset{\Delta}{=} \underset{r}{\arg\min}\, K(p_{expose}) \end{cases} \tag{3}$$

where $K(y) \overset{\Delta}{=} \dfrac{y''}{(1+y'^2)^{\frac{3}{2}}}$ represents the signed curvature of the curve $y = f(x)$.

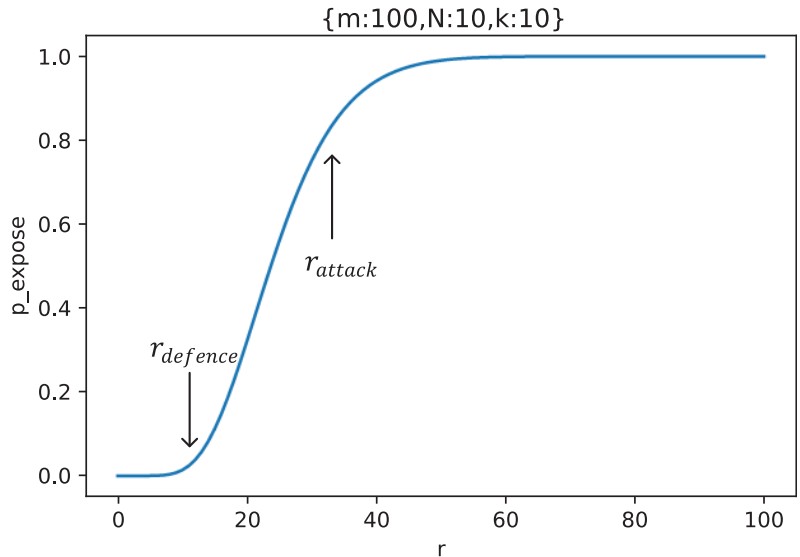

**Figure 8 The curve of key exposure probability $p_{expose}$ with the collusion number $r$ of malicious nodes ($m = 100, N = 10, k = 10$), where $r_{defense}$ is the upper limit of the defender's tolerance and $r_{attack}$ is the ideal lower limit of the number of malicious nodes controlled by the attacker.**

Since $\frac{\partial}{\partial r} p_{expose} = \frac{kN}{m} \left(1 - \frac{r}{m}\right)^{N-1} \left[1 - \left(1 - \frac{r}{m}\right)^N\right]^{k-1} \ll 1$, $r_{defense}$ and $r_{attack}$ can be approximated as

$$\begin{cases} r_{defense} \approx \arg\max_r \frac{\partial^2}{\partial r^2} p_{expose} \\ r_{attack} \approx \arg\min_r \frac{\partial^2}{\partial r^2} p_{expose} \end{cases} \quad (4)$$

Equation (4) can be obtained by solving equation $\frac{\partial^3}{\partial r^3} p_{expose} = 0$, so the analytical solutions of $r_{defense}$ and $r_{attack}$ are as follows:

$$\begin{cases} r_{defense} = \;<m\left[1 - \left(\frac{(N-1)(3Nk-N-4)+N\sqrt{\Delta}}{2(Nk-1)(Nk-2)}\right)^{\frac{1}{N}}\right]> \\ r_{attack} = \;<m\left[1 - \left(\frac{(N-1)(3Nk-N-4)-N\sqrt{\Delta}}{2(Nk-1)(Nk-2)}\right)^{\frac{1}{N}}\right]> \end{cases} \quad (5)$$

where $\Delta = (5Nk - N - k - 7)(N - 1)(k - 1)$ and $<x>$ represents the rounding of $x$.

Considering $r_{defense}$ and $r_{attack}$ increase linearly with $m$, let's define $\omega_{defense} \triangleq \frac{r_{defense}}{m}$ as the upper tolerance rate of the defender on the number of malicious nodes and $\omega_{attack} \triangleq \frac{r_{attack}}{m}$ as the ideal lower bound rate of the number of malicious nodes controlled by the attacker. The expressions of $\omega_{defense}$ and $\omega_{attack}$ are as follows:

$$\begin{cases} \omega_{defense} = 1 - \left(\frac{(N-1)(3Nk-N-4)+N\sqrt{\Delta}}{2(Nk-1)(Nk-2)}\right)^{\frac{1}{N}} \\ \omega_{attack} = 1 - \left(\frac{(N-1)(3Nk-N-4)-N\sqrt{\Delta}}{2(Nk-1)(Nk-2)}\right)^{\frac{1}{N}} \end{cases} \quad (6)$$

According to Eq. (6), we find that $\omega_{defense}$ and $\omega_{attack}$ are irrelevant to the number of middle forwarding nodes $m$, but only relevant to the number of forwarding times $N$ and the number of measured delays that make up the key $k$.

According to the expressions of $\omega_{defense}$ and $\omega_{attack}$, $\omega_{defense}$ has the following relationship with $\omega_{attack}$:

$$\begin{cases} (1-\omega_{defense})^N + (1-\omega_{attack})^N = \frac{(N-1)(3Nk-N-4)}{(Nk-1)(Nk-2)} \\ \\ (1-\omega_{defense})^N(1-\omega_{attack})^N = \frac{(N-1)(N-2)}{(Nk-1)(Nk-2)} \end{cases} \tag{7}$$

According to the **Vieta Theorem**, $(1-\omega_{defense})^N$ and $(1-\omega_{attack})^N$ can be regarded as the two roots of the characteristic equation described in Eq. (8).

$$x^2 - \frac{(N-1)(3Nk-N-4)}{(Nk-1)(Nk-2)}x + \frac{(N-1)(N-2)}{(Nk-1)(Nk-2)} = 0 \tag{8}$$

It is easier to solve $\omega_{defense}$ and $\omega_{attack}$ with the characteristic equation in an actual usage scenario. Take Fig. 8 as an example, in the scenario of setting $m = 100$, $N = 10$ and $k = 10$, the characteristic equation for solving $\omega_{defense}$ and $\omega_{attack}$ is

$$x^2 - \frac{143}{540}x + \frac{1}{135} = 0 \tag{9}$$

The two roots of Eq. (9) are $x_1 = (1-\omega_{defense})^{10} \approx 0.23353$ and $x_2 = (1-\omega_{attack})^{10} \approx 0.03178$, and $\omega_{defense} \approx 0.1354$ and $\omega_{attack} \approx 0.2917$ can be further solved. As $m = 100$, we can calculate exactly $r_{defense} = 14$ and $r_{attack} = 29$ in Fig. 8.

Furthermore, Fig. 9 shows the changing trend of $\omega_{defense}$ and $\omega_{attack}$ with the number of forwarding times $N$ and the number of delay measurement times $k$. Seen from the left figure of Fig. 9, $\omega_{defense}$ and $\omega_{attack}$ decrease with the increase of $N$. For the defender, the larger $\omega_{defense}$ and $\omega_{attack}$ are, the better the security is, so $N$ should not be set too large, and when $N$ is small ($N < 10$ in this figure), the change of $N$ has a great influence on $\omega_{defense}$ and $\omega_{attack}$. However, $N$ should not be set too large, when $N$ is large ($N > 20$ in this figure), there begins to have marginal effects. From the right figure of Fig. 9, $\omega_{defense}$ and $\omega_{attack}$ increase with the increase of $k$, and the influence decreases with the increase of $k$, and there is also a marginal effect. Compared with the numerical influence, $N$ has a greater influence on $\omega_{defense}$ and $\omega_{attack}$ than $k$.

Through analysis, we find that although Collusive Attack can destroy key privacy, a reasonable selection of deployment parameters $m, N, k$ can effectively improve the difficulty of the attacker's attack and reduce the risk of key exposure. Therefore, in Section 6, we will analyze effective defensive strategies against Collusive Attacks based on the attack-defense game.

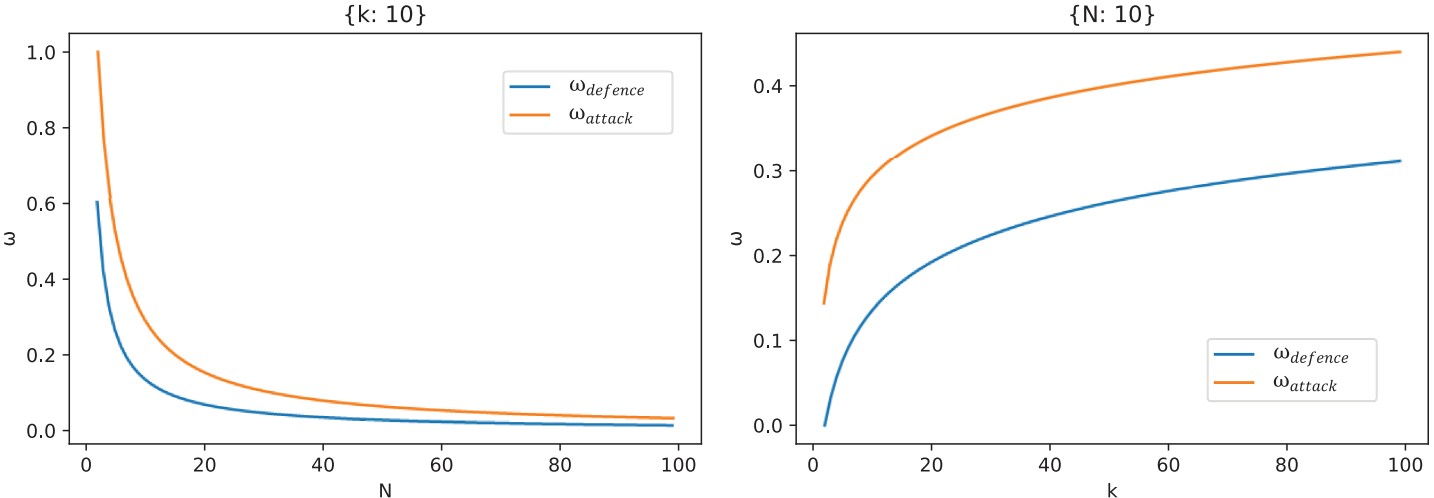

**Figure 9 The curves of $\omega_{defense}$ and $\omega_{attack}$ with forwarding times $N$ (when $k = 10$) and the number of measured delays $k$ (when $N = 10$).**

# SOLUTIONS ON THE SECURITY PROBLEM OF HUANG ET AL.'S SCHEME

## Anti-collusion attack defence strategy

According to the game theory, in the process of attack and defense confrontation, both sides will choose the most favorable strategy on the basis of the other side's strategy to achieve the goal of winning. And more importantly, the psychological expectations of attackers and defenders for network attacks are completely different, which will lead to different preferences of both sides in formulating strategies. Attackers usually take the attack cost into consideration when formulating attack strategies, because the core purpose of attackers is to make economic benefits. Defenders usually formulate defense strategies from the worst case, and this bottom-line thinking can guarantee the security of secret information to the greatest extent.

In the attack-defense game of Collusive Attack, the defender can select deployment parameters in advance, that is the number of middle forwarding nodes $m$, the random forwarding times $N$, and the number of delay measurement times $k$. While, according to the attack model, the status of the middle forwarding nodes is equal, and the attacker will randomly select the middle nodes to attack, so the attacker holds the variable $n$ of how many middle forwarding nodes to attack. In addition, because the attacker may fail to control the middle node, the attack success probability $\alpha \in [0, 1]$, which we called the out-of-control rate, is also an important parameter in the defense strategy, and it is decided by both the defender and the attacker.

For the defender, the defense strategy against collusion attack must be effective in the most dangerous environment, and the most dangerous environment is that the attacker adopts the optimal attack strategy to maximize the probability of key theft. What the defender is looking for is the deployment parameters that can still ensure the security of the

key when the attacker adopts the optimal attack strategy. Therefore, we divide the anti-collusion attack defense strategy into two main objectives:

1) For any given defense deployment parameters $m, N, k, \alpha$, finding the best attack strategy $n_{otp}$ makes the attacker's success probability $p_{success}$ the highest, which is to solve

$$n_{otp}(m, N, k, \alpha) = \arg\max_{n} p_{success}(n; m, N, k, \alpha) \qquad (10)$$

2) When the attacker adopts the best attack strategy $n_{otp}$, find the security deployment parameters $m^*, N^*, k^*, \alpha^*$ that make the attacker's optimal success attack probability $p_{success\_otp}$ close to 0, which is to solve

$$m^*, N^*, k^*, \alpha^* = \arg\min_{m,N,k,\alpha} p_{success}(m, N, k, \alpha; n_{otp}) \qquad (11)$$

In addition, considering the existence of different defense scenarios in reality, the expression of the attacker's probability of successful attack $p_{success}$ is different. According to the ability of the defender, we divide the defender into two scenarios: "Careless Defender" and "Cautious Defender". When the attacker fails to attack the middle forwarding node, the "Careless Defender" will not be alert, so the attacker can continue to attack. However, the "Cautious Defender" will alert the attacker and terminate the negotiation of the key after the attacker fails to control the attacked middle forwarding node.

In the centralized scenario, the middle forwarding nodes are usually managed by central control, which will monitor the status of the nodes. Therefore, once the attacker fails to control the middle forwarding nodes, the alarm will cause the central control to terminate the key agreement, so this scenario belongs to the "Cautious Defender" scenario. While in the decentralized scenario, the middle forwarding nodes are relatively independent and their defensive ability is usually weak, which makes it difficult for the attacker to be found even if the attack fails, so it is closer to the "Careless Defender" scenario.

But in either scenario, the secret apportionment strategy is the same. Therefore, in the game of attack and defense, the defender can take the following measures to minimize the $p_{success\_otp}$ against the attacker of the optimal attack strategy $n_{otp}$:

- Increase the number of middle forwarding nodes $m$. In the deployment of RFNs, whether it is a "Careless Defender" or a "Cautious Defender", increasing $m$ can achieve the purpose of reducing the probability of an attacker's successfully stealing the key. In the scenario of "Cautious Defender", increasing $m$ can have a significant effect, and a small-scale RFN can ensure sufficient security; While in the "Careless Defender" scenario, increasing $m$ has a less obvious effect on the success probability of the attacker. Fortunately, compared with the centralized "Cautious Defender" scenario, the decentralized "Careless Defender" scenario has a very low cost of increasing $m$.
- Appropriately reduces the forwarding times $N$. $N$ is closely related to the randomness of the measured delay, and a certain number of forwarding times $N$ can effectively improve the randomness of the measured delay, thus increasing the bit number of the key. However, too high $N$ will increase the probability of malicious nodes participating in

delay measurement, thus increasing the possibility of key leakage. Especially in the "Careless Defender" scenario, blindly increasing $N$ does great harm to key security; While in the "Cautious Defender" scenario, $N$ can be appropriately larger. Fortunately, unlike $m$, $N$ is a flexible and adjustable parameter of the defender, and the adjustment cost is low. The defender can dynamically choose an appropriate random forwarding number $N$ according to the security and randomness requirements.

- Appropriately increase the number of delay measurement times $k$. Because only the malicious node can obtain all the measured delays to crack the key, increasing the number of delay measurement times required for a single key can effectively reduce the risk of key leakage. Although the larger the number of delay measurement times $k$ is, the more secure the key is, it should not be too large, otherwise, the key generation rate will slow down. In fact, increasing the number of delay measurements times $k$ can quickly reduce the probability of key leakage, and it will soon be close to zero, whether in the scenario of "Careless Defender" or "Cautious Defender". In addition, $k$ is also a flexible and adjustable parameter of the defender like $N$. The defender can dynamically choose an appropriate number of delay measurement times $k$ according to the security and key generation rate requirements.

- Keep the out-of-control rate $\alpha$ of middle forwarding nodes at a low level. The out-of-control rate $\alpha$ is a very important parameter, representing how easy it is for the attacker to control the middle forwarding nodes. No matter in the scenario of "Careless Defender" or "Cautious Defender", once the middle forwarding node is completely out of control, in other words, $\alpha$ is close to 1, the probability of the attacker stealing the key under the optimal attack strategy $n_{otp}$ is close to 100% (this analysis ignores the economic costs of an attack). Therefore, the defender should take enough measures to reduce the out-of-control rate $\alpha$ of middle forwarding nodes and ensure that the nodes are secure and controllable. These tactics are well known, such as managing permissions for device visitors, setting up complex access keys, using firewalls, and patching operating system vulnerabilities. In addition, we find that "Cautious Defender" is much more tolerant of out-of-control rate $\alpha$ than "Careless Defender", because "Cautious Defender" had the supervision of middle forwarding nodes and detection of aggressive behavior. Therefore, in the scenario of "Careless Defender", it is also an important defense measure to find a decentralized alternative way to reach the effect of centralized supervision.

Next, we will prove the effectiveness of this defense strategy, and give an example of defense parameters that can resist collusion attacks.

## Proof of the effectiveness of our defense strategies

In this subsection, we will establish a probabilistic model for attack-defense game analysis in the scenarios of "Careless Defender" and "Cautious Defender" respectively to prove the effectiveness of the defense method. According to the two optimization objectives in previous subsection, the idea of proof is shown in Fig. 10, we will first solve the optimal attack strategy of the attacker under any deployment parameters, and then analyze the

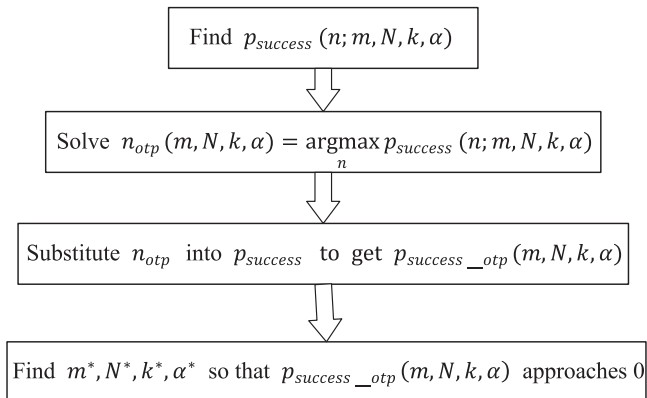

**Figure 10 Idea of proof.**               

relationship between the key theft probability and each deployment parameter under the optimal attack strategy of the attacker and the deployment parameter to ensure the security of the key. Finally, we will give an actual security deployment example.

### Careless defender

Since the attack behavior will not be found, assuming that the attacker chooses $n$ middle forwarding nodes as the attack targets, as long as there are middle forwarding nodes controlled by the attacker, a Collusive Attack can be launched to steal the key. Therefore, in the "Careless Defender" scenario, the probability $p_{success\_I}$ of the attacker successfully stealing the key is

$$p_{success\_I}(n) = \sum_{r=1}^{n} C_n^r \alpha^r (1-\alpha)^{n-r} \left[ 1 - \left( 1 - \frac{r}{m} \right)^N \right]^k \tag{12}$$

Figure 11 shows the change curve of attack success probability $p_{success\_I}$ with the number of attacking nodes $n$ under different out-of-control rate $\alpha$ when $m = 100$, $N = 10$ and $k = 10$.

From Fig. 11, we can see that $p_{success\_I}$ is a monotonic increasing function. Of course, generally, we need to prove that $p_{success\_I}(n + 1) > p_{success\_I}(n)$ (see Appendix A for proof).

The attacker hopes to find the optimal attack strategy in the "Careless Defender" scenario, that is, the optimal number of attack middle forwarding nodes $n_{I\_otp} \overset{\Delta}{=} \arg\max_{n} p_{success\_I}(n)$. As $p_{success\_I}$ is monotonically increasing, we find the optimal strategy is $n_{I\_otp} = m$. However, it is observed from Fig. 11 that when $\alpha$ is greater than a certain threshold, that is, $\alpha \geq \delta(m, N, k)$, the marginal effect will occur when the attacker increases the number of attack nodes, which is consistent with the meaning of the ideal lower limit $r_{attack}$ of the number of attack control nodes in the defense strategy analysis subsection of the defender. And when $\alpha \geq \delta(m, N, k)$, the optimal attack strategy should be $n_{attack}(\alpha) = \arg\max_{n} \frac{\partial^2}{\partial n^2} p_{success\_I}$ after taking the marginal effect into account, so overall, the optimal attack strategy in the "Careless Defender" scenario of the attacker is

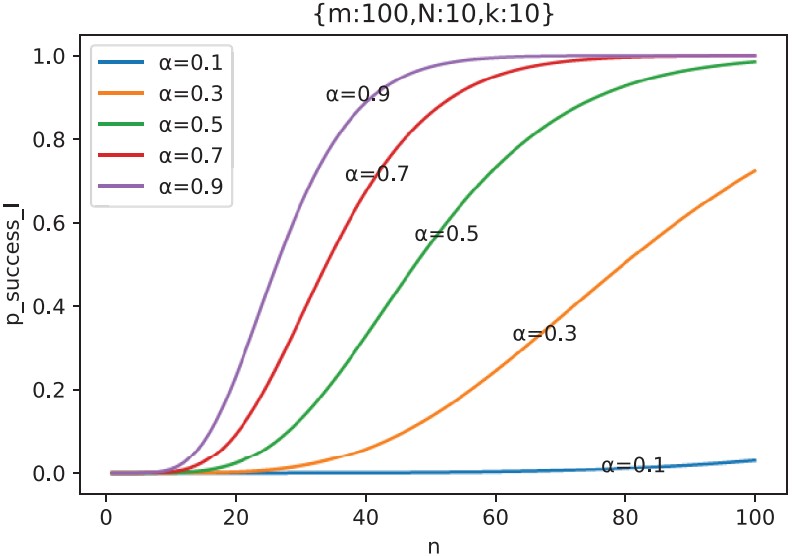

**Figure 11** The curve of attack success probability $p_{success\_I}$ in the scenario of "Careless Defender" with the number of attacking nodes $n$ under different out-of-control rate $\alpha$ ($m = 100, N = 10, k = 10$).

$$n_{I\_otp} = \begin{cases} m, \alpha < \delta(m, N, k) \\ n_{attack}(\alpha), \alpha \geq \delta(m, N, k) \end{cases}$$ (13)

It is difficult to solve $\arg \max \frac{\partial^2}{\partial n^2} p_{success\_I}$ directly because $p_{success\_I}$ has the discontinuous part $C_n^i$, so a approximate solution method is as follows.

**Lemma 1.** *For given $F(n)$, suppose $\exists G(n), G(n) \approx F(n)$. Then, for solution sets $Z = \{n | T\{F(n)\} = 0\}$ and $Z^* = \{n | T\{G(n)\} = 0\}$, we have $Z \approx Z^*$.*

*Proof.* Assume that $\varepsilon = \max |F(n) - G(n)|, G(n) \approx F(n)$ represents $\lim_{\varepsilon \to 0} G(n) = F(n)$. Then, we have $\lim_{\varepsilon \to 0} T\{G(n)\} = T\{F(n)\}$, so for $Z = \{n | T\{F(n)\} = 0\}$ and $Z^* = \{n | T\{G(n)\} = 0\}$, we have $\lim_{\varepsilon \to 0} Z^* = Z$, which is $Z \approx Z^*$.

To put it simply, if there exists $G(n)$ with good mathematical properties that can approximately replace $F(n)$ with bad mathematical properties, then any unsolvable mathematical equation about $F(n)$ can be replaced with $G(n)$ to solve the approximate solution.

It can be observed from Fig. 12 that the influence of $\alpha$ on $p_{success\_I}$ is actually approximately stretching the abscissa of the corresponding curve of $\alpha = 1$ (see Appendix B for proof), that is

$$\begin{aligned} p_{success\_I}(n; \alpha = \alpha_0) &\approx p_{success\_I}(\alpha_0 n; \alpha = 1) \\ &= p_{expose}(\alpha_0 n) \end{aligned}$$ (14)

Therefore, $p_{expose}(\alpha n)$ is an approximation of $p_{success\_I}$. According to Lemma. 1, we have

$$n_{attack}(\alpha) \approx \arg \max_n \frac{\partial^2}{\partial n^2} p_{expose}(\alpha n)$$ (15)

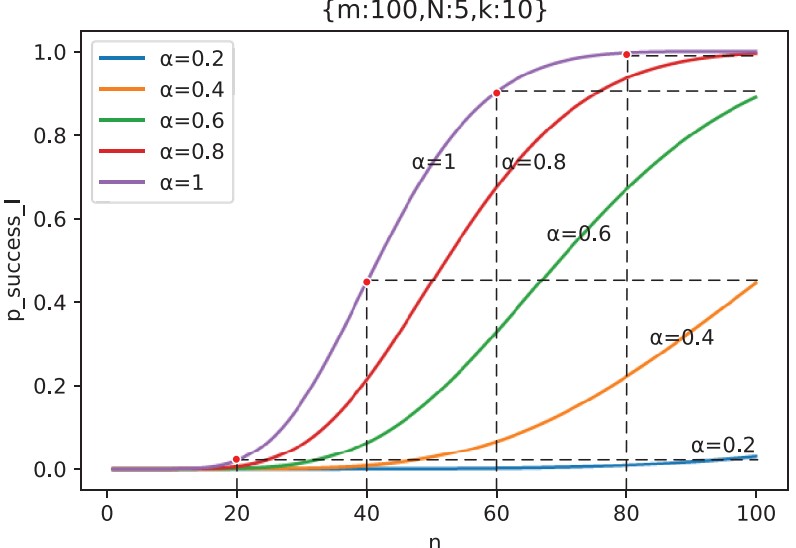

**Figure 12 The phenomenon of $p_{success\_1}$ under different out-of-control rate $\alpha$ being obtained by stretching along abscissa of curve with $\alpha = 1$ ($m = 100, N = 5, k = 10$).**

Since $\arg\max_n \frac{\partial^2}{\partial n^2} p_{expose}(n)$ has been solved in Eq. (5), $\arg\max_n \frac{\partial^2}{\partial n^2} p_{expose}(\alpha n)$ can be directly obtained by substitution method as

$$n_{attack}(\alpha) \approx < \frac{m}{\alpha} \omega_{attack} > \tag{16}$$

where $\omega_{attack}$ can be easily obtained by solving the characteristic equation described in Eq. (8) or directly obtained by Eq. (6).

As can be seen from Fig. 12, since $m$ is the upper bound of $n_{I\_otp}$, as long as $n_{attack}(\alpha) < m$, it means that marginal effect appears, then $n_{I\_otp} = n_{attack}(\alpha)$, otherwise $n_{I\_otp} = m$. So we can solve $n_{I\_otp}$ without knowing $\delta(m, N, k)$, which is

$$n_{I\_otp} = min[n_{attack}(\alpha), m] \tag{17}$$

Equation (17) gives the calculation formula of the attacker's optimal attack strategy $n_{I\_otp}$ in the scenario of "Careless Defender". It can be found that $\alpha$ has an important influence on the value of $n_{I\_otp}$, which is also of great interest to attackers. Therefore, we have made the curve of the optimal attack strategy $n_{I\_otp}$ of the attacker with the out-of-control rate $\alpha$ in the specific scenario shown in Fig. 13.

As can be seen from Fig. 13, in the scenario of "Careless Defender", when the out-of-control rate $\alpha$ of the middle forwarding node is small ($\alpha \leq \delta$), adding the middle forwarding nodes to attack will not cause a marginal effect, and the attacker's optimal strategy, in this case, is all-node attack.

However, when the out-of-control rate $\alpha$ of the middle forwarding nodes is large ($\alpha > \delta$), increasing the number of middle forwarding nodes in the attack will result in a marginal effect. Considering the attack income, the attacker's optimal strategy is to choose

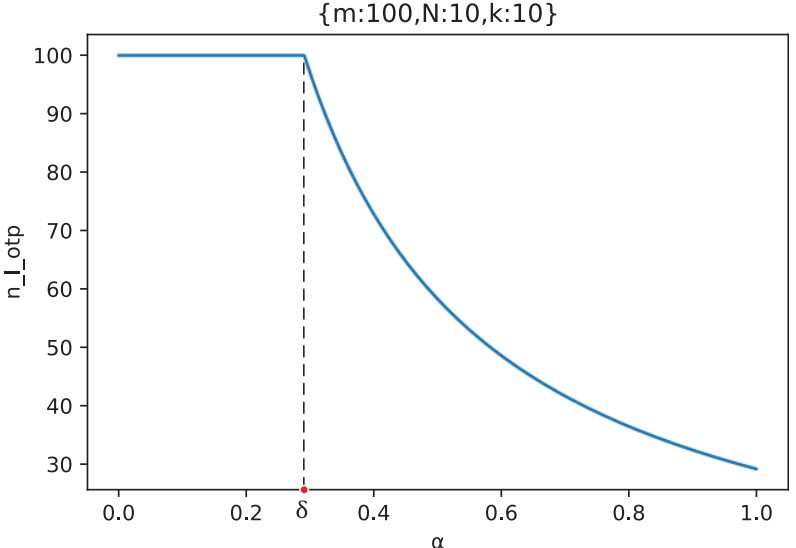

**Figure 13** The curve of the attacker's optimal attack strategy $n_{I\_otp}$ with the out-of-control rate $\alpha$ ($m = 100, N = 10, k = 10$).

the number of middle forwarding nodes before the appearance of the marginal effect, which is $n_{attack}(\alpha)$, and this number gradually decreases with the increase of $\alpha$.

On the whole, the larger the out-of-control rate $\alpha$ of the middle forwarding node is, the lower the number of middle forwarding nodes that the attacker may choose to attack according to the attacker's optimal strategy.

So, the probability $p_{success\_I\_otp}$ of the attacker successfully stealing the key under the optimal attack strategy $n_{I\_otp}$ in the scenario of "Careless Defender" is

$$p_{success\_I\_otp}(m, N, k, \alpha) \stackrel{\Delta}{=} p_{success\_I}(n_{I\_otp}) \tag{18}$$

Considering that the marginal part has little influence on the probability of the attacker's success, it is advisable to use $p_{success\_I}(m)$ to approximately describe $p_{success\_I}(n_{I\_otp})$, so we have

$$p_{success\_I\_otp}(m, N, k, \alpha) \approx p_{success\_I}(m) \tag{19}$$

Then, we explore the influence of $m, N, k, \alpha$ on $p_{success\_I\_otp}$ respectively. Figure 14 shows the relationship curves of the effects of four parameters in the typical case scenario.

It can be seen from Fig. 14 that $p_{success\_I\_otp}$ increases with the increase of the node out-of-control rate $\alpha$, and when $\alpha$ reaches a certain value ($\alpha > 0.5$ in this figure), the $p_{success\_I\_otp}$ approaches 100%, and the marginal effect appears; $p_{success\_I\_otp}$ increases with the increase of node forwarding times $N$. Similarly, when $N$ reaches a certain value ($N > 30$ in this figure), $p_{success\_I\_otp}$ approaches 100%, and marginal effect appears; $p_{success\_I\_otp}$ decreases with the increase of delay measurement times $k$. At the beginning of the increase, $p_{success\_I\_otp}$ drops rapidly, and when $k = 20$, $p_{success\_I\_otp}$ has dropped below 20% and continues to increase $k$, the decrease of $p_{success\_I\_otp}$ has a marginal effect; $p_{success\_I\_otp}$

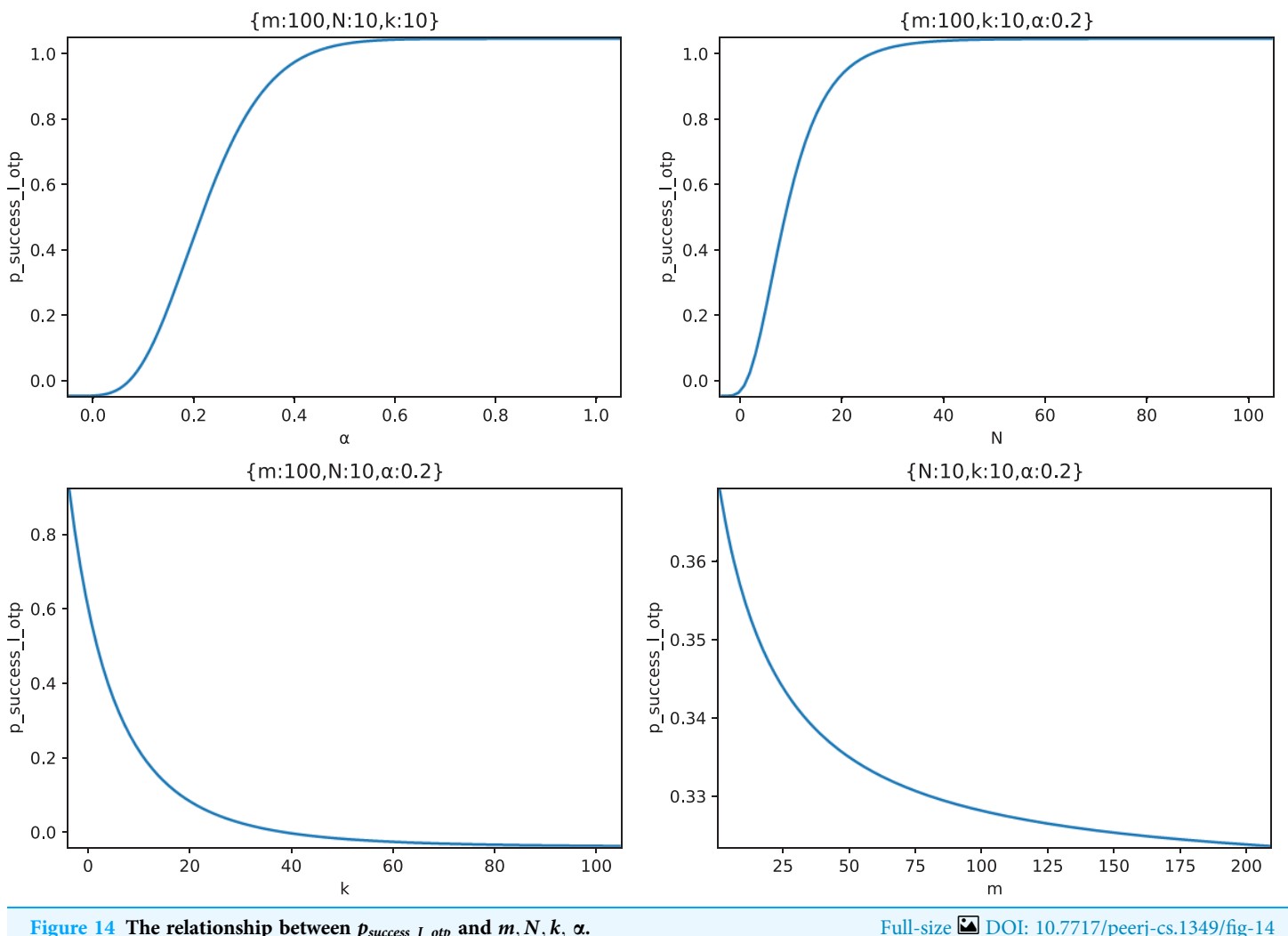

**Figure 14** The relationship between $p_{success\_1\_otp}$ and $m, N, k, \alpha$.

decreases with the increase of the number of middle forwarding nodes $m$. However, because the attacker adopts the optimal attack strategy, the impact of increasing $m$ on $p_{success\_1\_otp}$ is limited, and $m$ increased by 20 times in this figure only decreases $p_{success\_1\_otp}$ by 5%.

### Cautious defender

The "Cautious Defender" scenario means that there is a central control that judges whether there are abnormal middle forwarding nodes through traffic monitoring, behavior analysis, and other means. Once the abnormality is detected, the key agreement will be terminated in advance. Then, once the attacker attacks a certain middle forwarding node but fails to control it, it will trigger an alarm to alert the defender, resulting in the failure of stealing the key.

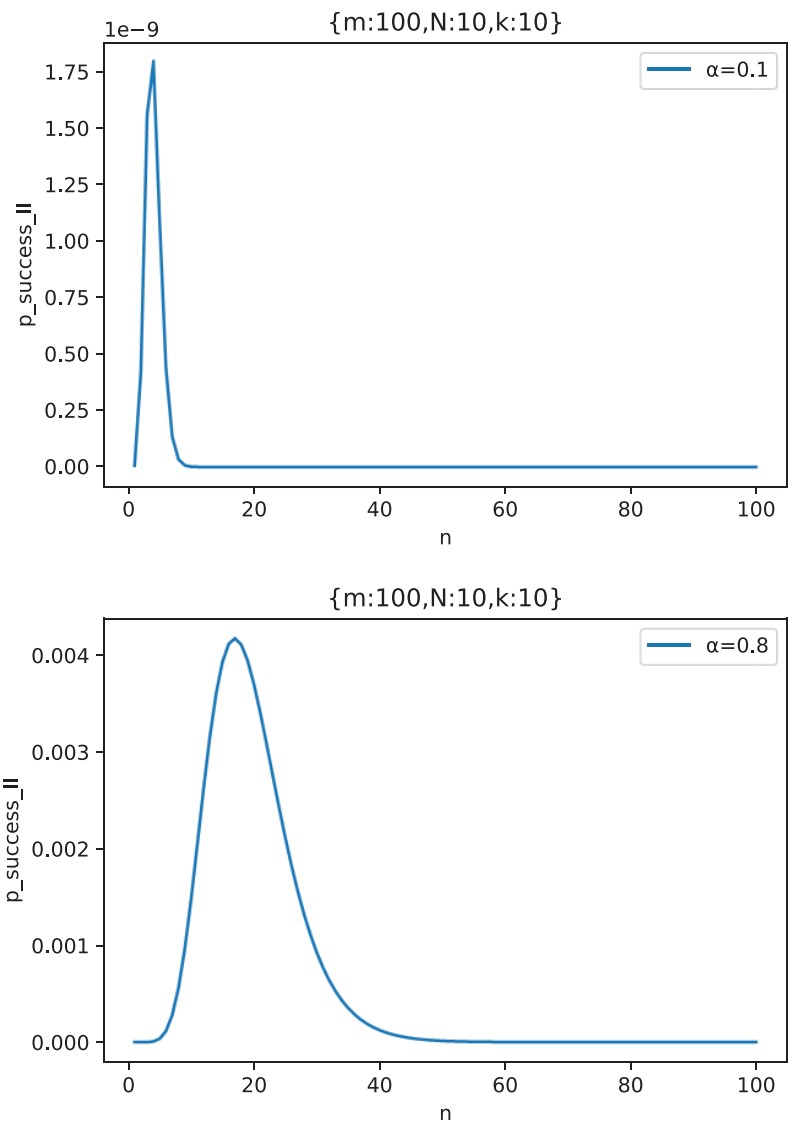

**Figure 15** The curve of attack success probability $p_{success\_I}$ in the scenario of "Cautious Defender" with the number of attacking nodes $n$ under two typical out-of-control rate $\alpha = 0.1, 0.8$ ($m = 100, N = 10, k = 10$).

Therefore, the attacker can not be found by the "Cautious Defender" only when all the targets are successfully controlled, so the probability $p_{success\_II}$ of the attacker successfully stealing the key in the "Cautious Defender" scenario is

$$p_{success\_II}(n) = \left[1 - \left(1 - \frac{n}{m}\right)^N\right]^k \alpha^n \tag{20}$$

Figure 15 shows the change curve of attack success probability $p_{success\_II}$ with the number of attacking nodes $n$ under two typical out-of-control rates $\alpha = 0.1, 0.8$ when $m = 100$, $N = 10$ and $k = 10$. We take two representatives $\alpha$ values, where $\alpha = 0.1$ indicates that the defense capability is strong and the attacker has not found effective defense vulnerabilities

and $\alpha = 0.8$ indicates that the defense capability is weak and the attacker has found effective defense vulnerabilities to attack.

From Fig. 15, we can see that $p_{success\_II}$ has a maximum value, that is, the attacker should choose $n_{II\_otp} \stackrel{\Delta}{=} \arg\max_{n} p_{success\_II}(n)$ number of middle forwarding nodes to attack, so as to maximize the success probability of the attack. The existence of this optimal solution is obvious. If the attackers choose too few middle forwarding nodes to attack, then even if all of them are controlled, they can not form an effective Collusive Attack, because the Collusive Attack requires a certain number of malicious nodes; If the number of nodes attacked by the attacker is too large, the probability of making a mistake on one middle forwarding node and triggering an alarm will be greater, which will also reduce the success rate of the attack. The rest of the $\alpha$ values have the same conclusion.

Therefore, only selecting an appropriate number of middle forwarding nodes to attack can make the attack more successful, which is also what the attacker hopes.

This optimal attack number $n_{II\_otp}$ can be obtained by solving $\frac{\partial}{\partial r} p_{success\_II} = 0$, which is the root of the equation described in Eq. (21).

$$L(n) = ln\alpha \left[ 1 - \left( 1 - \frac{n}{m} \right)^N \right] + \frac{kN}{m} \left( 1 - \frac{n}{m} \right)^{N-1} = 0 \tag{21}$$

Equation (21) is difficult to solve directly. However, considering that "Cautious Defender" commonly chooses more middle forwarding nodes to keep the key secure and that the out-of-control rate $\alpha$ of these middle forwarding nodes is small under the tight defense strategy of the "Cautious Defender", which causes $\frac{n}{m} \to 0$ according to Fig. 15. Therefore, we have the following approximation:

$$\left( 1 - \frac{n}{m} \right)^N \approx 1 - \frac{Nn}{m} \tag{22}$$

By substituting Eq. (22) into Eq. (21), we get

$$n_{II\_otp} \approx \left< \frac{mk}{k(N-1) - mln\alpha} \right>, \alpha \leq \mu \tag{23}$$

where $\mu$ is an empirical boundary that satisfies this approximation.

When $\alpha$ becomes larger ($\alpha > \mu$), because Eq. (21) is a transcendental equation, the analytical solution cannot be obtained directly. It also can not be approximated by Eq. (22), but the approximate numerical solution can be obtained by Newton's Method $n_t = n_{t-1} - \frac{L(n_{t-1})}{L'(n_{t-1})}$, and the initial value can be chosen as $n_0 = 1$. Then, we have

$$n_{II\_otp} \approx \lim_{t \to +\infty} n_t, \alpha > \mu \tag{24}$$

Taken together from Eqs. (23) and (24), the optimal attack strategy $n_{II\_otp}$ of the attacker in the "Cautious Defender" scenario is

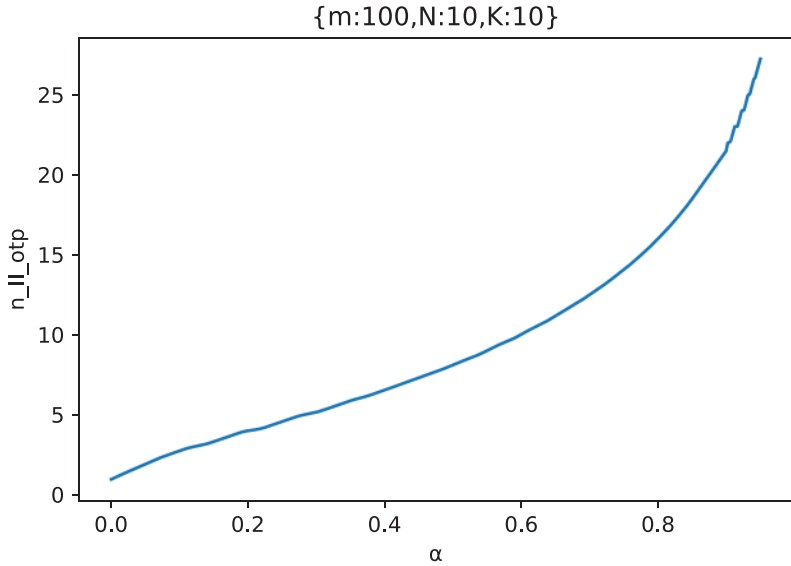

**Figure 16 The curve of the attacker's optimal attack strategy $n_{II\_otp}$ with the out-of-control rate $\alpha$ ($m = 100, N = 10, k = 10$).**

$$n_{II\_otp} = \begin{cases} < \frac{mk}{k(N-1)-mln\alpha} >, \alpha \leq \mu \\ \lim_{t \to +\infty} n_t, \alpha > \mu \end{cases} \tag{25}$$

From the experimental results (see Appendix C), it is appropriate to take about $\mu \approx 0.2$. This is a relatively conservative estimate. In addition, $\alpha \leq 0.2$ is usually satisfied in the "Cautious Defender" scenario, and the calculation of the optimal attack strategy of the attacker can be solved directly by the simple formula method, and Newton's method is seldom used.

Figure 16 shows the curve of the optimal attack strategy $n_{II\_otp}$ of the attacker with the out-of-control rate $\alpha$ in an example scenario.

It can be seen from Fig. 16 that as the out-of-control rate $\alpha$ increases, the attacker should gradually increase the number of attacking middle forwarding nodes. In addition, it can also be found that when $\alpha$ is large enough, although the attack success rate is very high, the attacker should not fully attack all middle forwarding nodes, but choose the appropriate number of middle forwarding nodes to attack.

The probability $p_{success\_II\_otp}$ of the attacker successfully stealing the key under the optimal attack strategy $n_{II\_otp}$ in the scenario of "Cautious Defender" is

$$p_{success\_II\_otp}(m, N, k, \alpha) \overset{\Delta}{=} p_{success\_II}(n_{II\_otp}) \tag{26}$$

Similarly, we draw Fig. 17 to show the relationship curves of the effects of four parameters $m, N, k, \alpha$ on $p_{success\_II\_otp}$ in the typical case scenario.

It can be seen from Fig. 17 that $p_{success\_II\_otp}$ increases with the increase of the node out-of-control rate $\alpha$, and only when $\alpha$ reaches a certain value ($\alpha > 0.8$ in this figure) does $p_{success\_II\_otp}$ begin to increase significantly; $p_{success\_II\_otp}$ increases with the increase of node

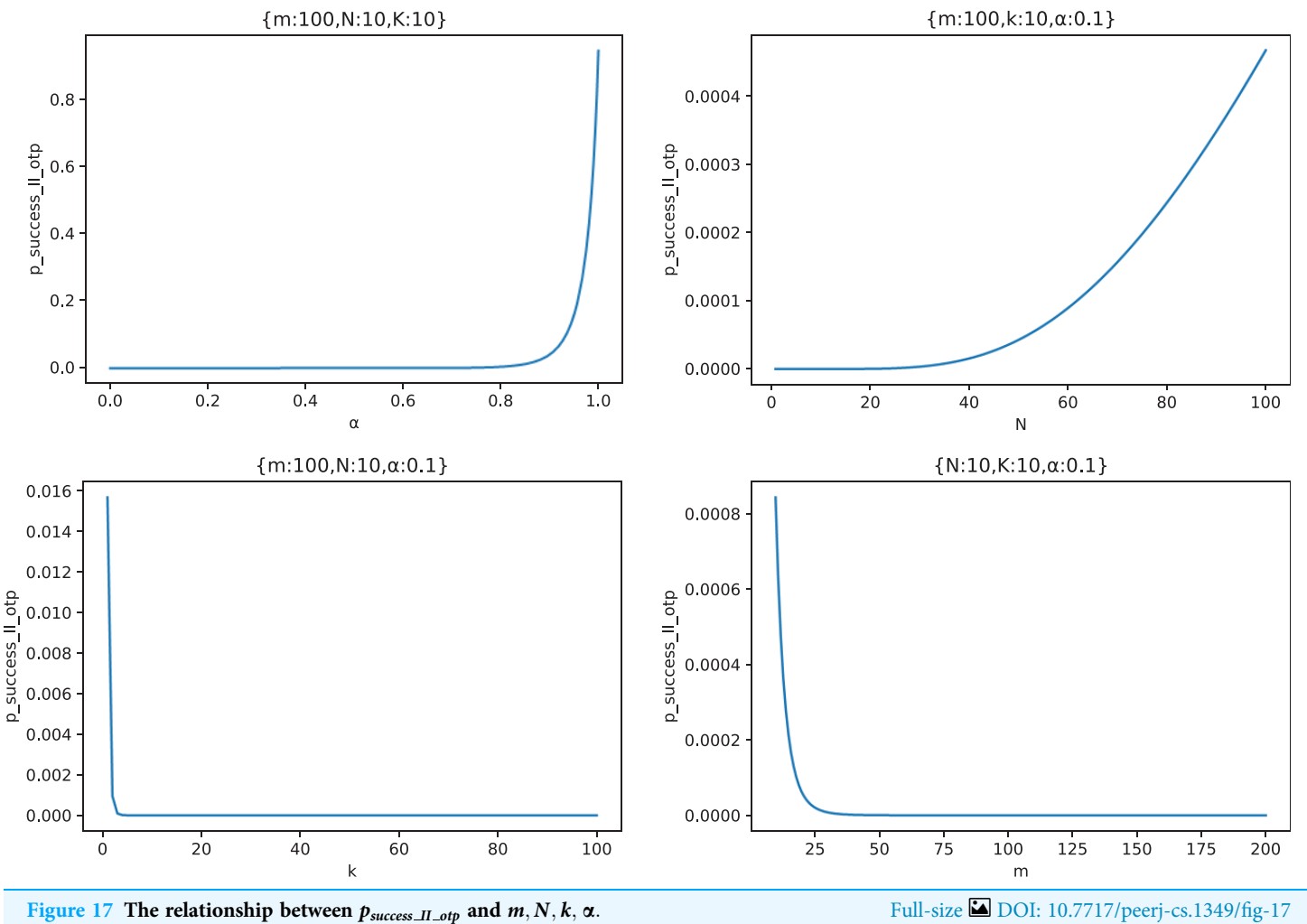

**Figure 17** The relationship between $p_{success\_II\_otp}$ and $m, N, k, \alpha$.

forwarding times $N$. Similarly, when $N$ reaches a certain value ($N > 40$ in this figure), $p_{success\_II\_otp}$ begins to increase significantly; $p_{success\_II\_otp}$ decreases with the increase of delay measurement times $k$. At the beginning of the increase, $p_{success\_II\_otp}$ drops rapidly, and $p_{success\_II\_otp}$ has basically dropped to 0 when $k > 4$ in this figure; $p_{success\_II\_otp}$ decreases with the increase of the number of middle forwarding nodes $m$. When $m$ reaches a certain value ($m > 25$ in this figure), $p_{success\_II\_otp}$ has basically dropped to 0.

From the analysis of the attack-defense game in the two different scenarios, it can be clearly seen that the defenders are dominant in the whole game. Because the defender can control most parameters $m, N, k$, and each parameter has a significant impact on the attacker's attack success rate $p_{success\_otp}$ under the best strategy. Generally speaking, as long as we select deployment parameters according to our security defense strategy, even if the attacker chooses the optimal strategy, the defender can also reduce the attack success probability to almost zero, so that the security of the physical layer key generation scheme

based on measured delay against collusion attack can achieve absolute security in the sense of probability.

For example, if a "Careless Defender" uses 100 middle nodes with the out-of-control rate of 0.2 to form a random forwarding network, the security of the key can be guaranteed by setting the forwarding times to 10 and the key to be composed of 40 measured delays. Because it can be proved from Fig. 14 that $p_{success\_I\_otp} \approx 0$ under the parameters that are $m = 100, N = 10, k = 40, \alpha = 0.2$. The same goes for a "Cautious Defender".

## LITERATURE REVIEW

With the advent of the IoT era, encryption methods based on traditional cryptography cannot cover a huge number of smart devices. However, encryption methods based on PLS can realize lightweight encryption and decryption, which is the first choice to ensure the communication security of IoT devices.

In 1949, *Shannon (1949)* gave the definition of perfect secrecy and proved the unconditional security of one secret at a time by using two theorems about perfect secrecy. On this basis, *Wyner (1975)* put forward a mathematical model of an eavesdropping channel, assuming that the eavesdropping channel is the degraded channel of the legitimate receiver. *Maurer (1993)* showed that correlation randomness can be used to generate keys and thought that Wyner's degraded eavesdropping channel may not be realistic, and proposed a key agreement protocol in which both parties can communicate securely. The key elements of this scheme are information reconciliation and security enhancement (*Maurer, 1993*).

Scholars continued Maurer's idea and carried out research in the field of PLKG. In the field of wireless communication, with the development of 5G mobile communication technology, key generation technologies based on the single antenna (*Abbasi et al., 2020*) and MIMO have been proposed (*Melki et al., 2020*). Channel characteristics such as received signal strength (RSS), channel characteristic information (CSI), and angle of arrival (AOA) have been proven to be applicable to key generation, while MIMO technology can effectively improve the problem of low secure key generation rate (SKGR) due to single channel (*Jiao, Tang & Zeng, 2018*). In recent years, with the rise of a breakthrough wireless communication technology-Reconfigurable Intelligent Surface (RIS) (*Shengjie et al., 2022*), RIS can be used to synthesize high-entropy dynamic channels under the uncontrollable wireless environment, which is considered to be an effective solution to the problems of poor reliability and difficult key generation caused by the traditional wireless channel physical layer key generation technology in the face of harsh communication environment (*Li et al., 2021*). In addition to the channel characteristics suitable for key generation in wireless communication, scholars have also designed PLKG methods in fields including visible light communication, underwater communication, and wired communication, which greatly enriches the application scenarios of PLKG. Of course, the most special method is the PLKG method based on transmission delay proposed by *Huang et al. (2021)* which uses the physical characteristics of the network itself to establish secure communication, gets rid of the restrictions on various

communication modes, and can be well compatible with all current networks, and has a wide range of application prospects.

In addition to the mining of available channel features, the research on specific key generation technology has also attracted the attention of many scholars. The typical processes to obtain the shared secret key include channel measurement, quantization coding, information reconciliation, and privacy amplification (*Shehadeh & Hogrefe, 2015*). Through these processes, the randomness of the reciprocal channel characteristics is completely retained in the secret key to the maximum extent, and at the same time, the minimum information leakage is also a key concern. The main purpose of quantization is to discretize the random channel measurement values and to preserve the randomness of channel measurement values to the greatest extent while removing some noise through reasonable quantization order and quantization interval settings. Therefore, in the face of different channel characteristics, it has a very important influence on the key bit rate to design a quantization scheme that matches the distribution of the characteristics. At the same time, the quantization coding process also needs to ensure that both parties have a high agreement rate of key bits for information reconciliation (*Wu, Xia & Cheng, 2018*). The essence of information reconciliation is to correct the inconsistencies by channel coding. Famous information reconciliation protocols include Binary, Cascade, and Winnow protocol which combines checksum and Hamming code for information reconciliation. In recent years, information reconciliation schemes based on LDPC code and Polar code have also been proposed (*Zhang et al., 2018*). In order to further reduce the impact of leaked information on key security in the process of information reconciliation, Bennett, Brassard and Robert introduced the concept of privacy amplification for special situations, extracting highly confidential secrets from a large number of shared information to generate keys (*Bennett et al., 1995*).

Although the research of PLKG has achieved great success, there are still many problems in practical application. On the one hand, it is difficult to ensure that the randomness of channel characteristics can meet the high key generation rate required by the application, on the other hand, there are potential security problems to be solved, including collusion attacks, so the research of PLKG still has a long way to go.

## CONCLUSIONS

*Huang et al. (2021)* propose an innovation key agreement method based on network physical features to solve the secure communication problem of large-scale heterogeneous devices. The method uses the RFNs and three-stage delay measurement protocol to generate reciprocal random measured delays. Communication parties can utilize these measured delays to share the key through quantization coding and information reconciliation. We study the security mechanism of this method and discover that the security of our method is based on the Secret Apportionment Strategy, where the main security threat comes from the Collusive Attack. We deduce the influence of the Collusive Attack on Secret Apportionment Strategy through game theory and give the best defense strategy for the defender to ensure key security.

### Funding

This work was supported by the National Key Research and Development Program of China (No. 2018YFB2100403). The funders had no role in study design, data collection and analysis, decision to publish, or preparation of the manuscript.

### Grant Disclosures

The following grant information was disclosed by the authors:
National Key Research and Development Program of China: 2018YFB2100403.

### Competing Interests

The authors declare that they have no competing interests.

### Author Contributions

- Xiaowen Wang conceived and designed the experiments, performed the experiments, analyzed the data, performed the computation work, prepared figures and/or tables, authored or reviewed drafts of the article, and approved the final draft.
- Jie Huang conceived and designed the experiments, performed the experiments, analyzed the data, performed the computation work, prepared figures and/or tables, authored or reviewed drafts of the article, and approved the final draft.
- Chunyang Qi conceived and designed the experiments, performed the experiments, analyzed the data, performed the computation work, prepared figures and/or tables, authored or reviewed drafts of the article, and approved the final draft.
- Yang Peng conceived and designed the experiments, performed the experiments, analyzed the data, performed the computation work, prepared figures and/or tables, authored or reviewed drafts of the article, and approved the final draft.
- Shuaishuai Zhang conceived and designed the experiments, performed the experiments, analyzed the data, performed the computation work, prepared figures and/or tables, authored or reviewed drafts of the article, and approved the final draft.

### Data Availability

Figures 8–18 in the article can all be generated from the code files.

### Supplemental Information

Supplemental information for this article can be found online at http://dx.doi.org/10.7717/peerj-cs.1349#supplemental-information.

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
