# Peer review of "An anti-collusion attack defense method for physical layer key generation scheme based on transmission delay"

_PeerJ Computer Science, doi:10.7717/peerj-cs.1349_

## Round 0.1 · original submission · Minor Revisions

Authors are advised to consider the comments of all reviewers carefully and consider proofreading carefully.

Reviewer 1 ·

Basic reporting

The article have sufficient introduction and background to demonstrate how the work fits into the broader field of knowledge. Relevant prior literature review is discussed properly.

Author should revise his Abstract and add highlight his contribution.

Experimental design

The submission has defined proper research question with appropriate purpose.
The knowledge gap has been investigated, identified, and statements are clear for reader.
However, I suggest author to extend the Introduction part and add the Literature Review heading separately to add the research gap.

Validity of the findings

The data on which the conclusions are based is provide properly.

Additional comments

Revise the paper flow and highlight your contribution in Abstract.

Reviewer 2 ·

Basic reporting

The paper titled "An anti-collusion attack defense method for physical layer key generation scheme based on transmission delay", is a novel writing as so far there are articles on anti-collusion attack defense method exist but for anti-collusion attack defense method for physical layer key generation scheme based on transmission delay is rare. The use of the influence of the number of conspiring malicious nodes on the exposure probability of the key generated by this scheme through the probability model is very impressive. The paper is clearly written in a good style and includes figures and tables wherever necessary.

Experimental design

The objective and motivation for the research has been very well stated in the introduction part. But needs clarification on the following:
1. Author divide the anti-collusion attack defense strategy into two main objectives(In section SOLUTIONS ON THE SECURITY PROBLEM OF HUANG ET AL.’S SCHEME, These objective need more clarification

Validity of the findings

Use of probabilistic models in the scenarios of “Careless Defender” and “Cautious Defender” respectively to prove the effectiveness of the defense method is quite satisfactory. The authors have clearly acknowledged and identified the contributions of their research against previous researchers' work..

The authors adequately evaluated their work, and all claims are clearly articulated and supported by empirical experiments

Additional comments

However, addressing the above comments would improve the quality of the paper. The overall work is good, novel and timely.

Reviewer 3 ·

Basic reporting

The article aims fill the gap at the Hung et al.’s scheme. The authors stated that they find a security problem of collusion attack. It is better to start the abstract (line#12) by identifying the research gap (research problem) instead of identifying the reference.
At (line#57), Huang et al.’s scheme is the main reference of the research gap at the article, why this reference is selected among others? Do you think it is the main and the only reference at this domain? At (line#66), the author proposed an anti-collusion attach defense method, what is the unique at the proposed method here? Since it has been also discussed by previous studies.
At (line#352), required to update the introduction to reflect the goals of the current phase, and also required to numbering the sub-sections that under the main topic.
The authors analyze the research problem with updated references that matched with the research problem. Suggested to add show the research process through a flowchart or some illustrated steps to be clear for the reader.

Experimental design

'no comment'

Validity of the findings

The author stated at (line#20), that the "Through theoretical analysis, we prove the
effectiveness of the anti-collusion defence method proposed ".
Need to discuss the practical approach that can be used to validate the findings also, since the authors consider only the theoretical analysis at a based to confirm the validity of the suggested approach.

Additional comments

Overall, the research topic and writing with the analysis is very good.
English proofreading is needed for the article.

---

## Round 0.2 · accepted · Accept

Congratulations to the authors on acceptance. Thanks for your contribution.

Reviewer 2 ·

Basic reporting

The paper titled "An anti-collusion attack defense method for physical layer key generation scheme based on transmission delay", is a novel writing as so far there are articles on anti-collusion attack defense method exist but for anti-collusion attack defense method for physical layer key generation scheme based on transmission delay is rare. The use of the influence of the number of conspiring malicious nodes on the exposure probability of the key generated by this scheme through the probability model is very impressive. The paper is clearly written in a good style and includes figures and tables wherever necessary.

Experimental design

The objective and motivation for the research has been very well stated in the introduction part. The authors clarified the following issue raised by reviewer in first review .
1. Author divide the anti-collusion attack defense strategy into two main objectives(In section SOLUTIONS ON THE SECURITY PROBLEM OF HUANG ET AL.’S SCHEME, These objective need more clarification -
Use of probabilistic models in the scenarios of “Careless Defender” and “Cautious Defender” respectively to prove the effectiveness of the defense method is quite satisfactory. The authors have clearly acknowledged and identified the contributions of their research against previous researchers' work.

Validity of the findings

Validity of findings:
The authors adequately evaluated their work, and all claims are clearly articulated and supported by empirical experiments.

Additional comments

The overall work is good, novel and timely.

Reviewer 3 ·

Basic reporting

The authors improved the article with consideration of the comments.

Experimental design

No comment

Validity of the findings

No comment